# Fermionic quantum computation with Cooper pair splitters

Kostas Vilkelis[1*, 2], Antonio Manesco[2†], Juan Daniel Torres Luna[1, 2], Sebastian Miles[1, 2], Michael Wimmer[1, 2], Anton Akhmerov[2]

**1** Qutech, Delft University of Technology, Delft 2600 GA, The Netherlands
**2** Kavli Institute of Nanoscience, Delft University of Technology, Delft 2600 GA, The Netherlands
* kostasvilkelis@gmail.com, †am@antoniomanesco.org

June 6, 2024

## Abstract

We propose a practical implementation of a universal quantum computer that uses local fermionic modes (LFM) rather than qubits. The device consists of quantum dots tunnel-coupled by a hybrid superconducting island and a tunable capacitive coupling between the dots. We show that coherent control of Cooper pair splitting, elastic cotunneling, and Coulomb interactions implements the universal set of quantum gates defined by Bravyi and Kitaev [1]. Due to the similarity with charge qubits, we expect charge noise to be the main source of decoherence. For this reason, we also consider an alternative design where the quantum dots have tunable coupling to the superconductor. In this second device design, we show that there is a sweet spot for which the local fermionic modes are charge neutral, making the device insensitive to charge noise effects. Finally, we compare both designs and their experimental limitations and suggest future efforts to overcome them.

**See also:** online presentation recording

# 1 Introduction

Over the years, qubits emerged as the de facto basis for quantum computation with a plethora of host platforms: superconducting circuits [2,3], trapped ions [4,5] and quantum dots [6], to name a few. Recent works used qubit-based quantum computers to simulate fermionic systems [7–9]. However, the mapping from qubits to local fermionic modes (LFMs) is inefficient because it introduces additional overhead to the calculations [10, 11]. For example, a map from $n$ qubits to fermions requires $O(n)$ additional operations through the Jordan-Wigner transformation [12] and $O(\log n)$ through the Bravyi-Kitaev transformation [1].

An alternative to avoid the overhead in the qubit to LFM map is to use a quantum computer that already operates with local fermionic modes [1]. Moreover, the advantage of local fermionic modes is not limited to fermionic systems simulations. A set of $2n$ local fermionic modes maps directly to $n$ parity-preserving qubits or $n-1$ qubits. Therefore, the map from local fermionic modes to qubits only requires a constant number of operations regardless of the system size, being more efficient than the inverse [1]. Recently, Refs. [13, 14] showed that local fermionic modes offer advantages in quantum optimization problems of finding the ground state energy of fermionic Hamiltonians.

There exist several proposed platforms to implement fermionic quantum computation. Reference [15] encodes LFMs into noise-protected Majorana modes and implements gate operations through a combination of braiding and rotations. However, Majorana modes are still elusive, and braiding operations remain an experimental challenge. Recently, Ref. [16] proposed neutral atoms confined by and manipulated through optical tweezers as another LFM platform. The neutral atoms platform offers high-fidelity gates and coherence times above the millisecond range but suffers from scalability issues and slow gate operations with characteristic times of 1–100µs [4, 17]. We propose an alternative solid-state platform for fermionic quantum computation. Our proposal is inspired by recently reported advances in Cooper pair splitters [18–27]. The design includes an additional tunable capacitance to control interdot interactions. We show that the device implements the necessary universal set of gates proposed by Bravyi and Kitaev [1]. We also discuss the limitations of the device.

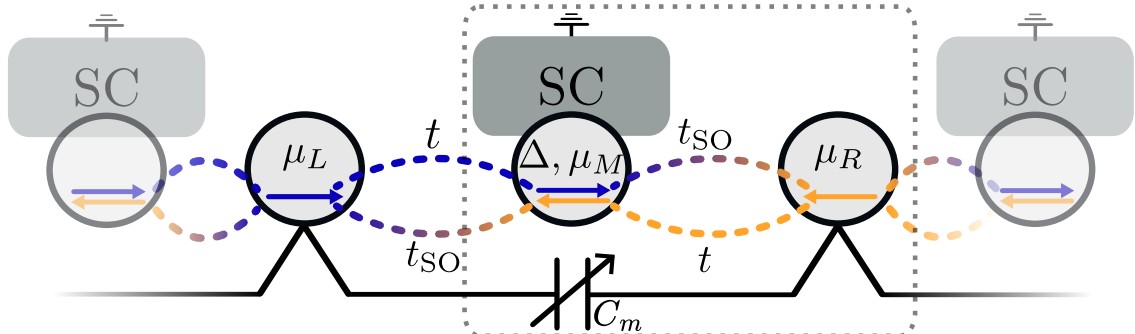

Figure 1: Minimal device implementation for universal fermionic computation. The unit cell of a fermionic processor—a fermion and a coupler—is indicated by the dashed grey box. For operations, two singly-occupied spin-(anti)polarized quantum dots host the local fermionic modes $L$ and $R$. Two tunnel barriers enable normal $t$ and spin-dependent $t_{SO}$ tunnelings between the two dots. A middle superconducting island mediates superconducting correlations between the two local fermionic modes. An external mutual capacitor $C_m$ allows Coulomb interactions between the sites.

## 2   Design

Bravyi and Kitaev [1] showed that fermionic quantum computation is equivalent to parity-preserving qubit operations. As a consequence, given a set of fermionic creation ($c_i^\dagger$) and annihilation operators ($c_i$), it follows that

$$\left\{ \begin{array}{ll} \mathcal{U}_1(\alpha) = \exp\left(i\alpha c_i^\dagger c_i\right) \;, & \mathcal{U}_2(\beta) = \exp\left[i\beta\left(c_i^\dagger c_j + c_j^\dagger c_i\right)\right] \;, \\[2mm] \mathcal{U}_3(\gamma) = \exp\left[i\gamma\left(c_i^\dagger c_j^\dagger + c_j c_i\right)\right] \;, & \mathcal{U}_4(\delta) = \exp\left(i\delta c_i^\dagger c_i c_j^\dagger c_j\right) \end{array} \right\} \tag{1}$$

with $\alpha = \beta = \gamma = \pi/4$, and $\delta = \pi$, is a universal set of gate operators. The case of two LFMs is similar to two uncoupled qubits: each operation within a given fermion parity sector is a rotation within SU(2). In the odd fermion parity sector, the operations $\mathcal{U}_1(\alpha)$ and $\mathcal{U}_2(\beta)$ are rotations around perpendicular axes in the Bloch sphere. Likewise $\mathcal{U}_3(\gamma)$ and $\mathcal{U}_4(\delta)$ are perpendicular rotations within the even fermion parity sector. The $\mathcal{U}_4$ operation acts as a CZ two-fermion gate. In the presence of extra LFMs, required to create superpositions of joint fermion parity of LFMs $i$ and $j$, applying $\mathcal{U}_4$ generates states with multi-fermion entanglement.

We thus propose a device where excitations occupy single-orbital sites, numbered by the subindex $i$ and $j$. A possible platform for such a proposal is an array of spin-polarized quantum dots, as the scheme shown in Fig. 1. Within this platform, the unitary operations in Eq. 1 are a time-evolution of the following processes:

1. $c_i^\dagger c_i$ onsite energy shift of the fermionic state at site $i$;

2. $c_i^\dagger c_j$ hopping of a fermion between sites $i$ and $j$;

3. $c_i^\dagger c_j^\dagger$ superconducting pairing between fermions at sites $i$ and $j$;

4. $c_i^\dagger c_i c_j^\dagger c_j$ Coulomb interaction between fermions at sites $i$ and $j$.

We control the onsite energies $\mu_i$ with plunger gates. Similarly, a tunnel gate between neighboring pairs of quantum dots controls hopping strength $t$ between them. Manipulation with plunger and tunnel gates is a well-established technique in charge [28, 29] and spin [6] qubits.

To implement the superconducting coupling between the spin-polarized dots, we utilize the design of a triplet Copper pair splitter [19–23, 27]. We include an auxiliary quantum dot in proximity to an s-wave superconductor mediating crossed Andreev reflection (CAR) and elastic cotunelling (ECT) between the two quantum dots that encode the LFMs. Thus, the ECT rate $\Gamma$ sets the hopping strength between the two dots, whereas the CAR rate $\Lambda$ sets the effective superconducting pairing. Because the dots are spin-polarised, the superconducting pairing must be of spin-triplet type, enabled by spin-orbit hopping in the hosting material. We quantify the spin-orbit coupling in the hosting material by the spin precession angle between the dots $\theta_i = 2\pi d/l_{so}$, where $d$ is the interdot distance and $l_{so}$ is the spin-orbit length. The spin-orbit coupling in InSb wires leads to a spin-precession length $l_{\rm so} \geq 100\,{\rm nm}$ [30–32] resulting in non-negligible $\theta_i$ within the order of the dot-to-dot distance.

Finally, we achieve Coulomb interaction between a pair of dots through capacitive coupling $C_m$. Our design requires a variable capacitive coupling to implement the $\mathcal{U}_4$ gate. Several recent works demonstrate variable capacitive coupling in various platforms: superconducting islands with variable Josephson energy [33], external double quantum dots [34], gate-tunable two-dimensional electron gas [35] and varactor diodes [36].

We show a minimal design of a fermionic quantum computer with two LFMs in Fig. 1. The device consists of three tunnel-coupled quantum dots in a material with large spin-orbit coupling. The middle dot is proximitized by an s-wave superconductor with an induced gap $\Delta$ that mediates CAR and ECT between the outer dots. The spin-polarised outer dots $(L, R)$ encode the LFMs, whereas the middle one is an auxiliary component. Finally, a tunable capacitor couples the outer dots. We generalize the device to an arbitrary number of LFMs by repeating the unit cell indicated by the grey dashed box in Fig. 1 in a chain. To read out the fermionic state, we propose to measure the occupation in each quantum dot through charge sensing [37].

## 3 Effective Hamiltonian

### 3.1 Single fermion processes

In the absence of capacitive and tunnel coupling, the approximate Hamiltonian for the two spin-polarised dots is

$$H_d = \sum_{i=L,R} \mu_i c^\dagger_{i\sigma_i} c_{i\sigma_i} \;, \tag{2}$$

where $c_{i\sigma}$ is the electron anihilation operator at site $i$ and spin $\sigma$ while $\mu_i$ is the corresponding chemical potential. The Hamiltonian in Eq. (2) is valid under two conditions: (i) the Zeeman splitting is sufficiently large to ensure the spin polarisation; (ii) because $\mu_i$ is a tunable parameter and the singlet state has no Zeeman splitting contribution, the charging energy must be sufficiently large to ensure that doubly-occupied states are well-separated from the computational states. Recent experiments on similar devices measure charging energy of $2\,{\rm meV}$ and Zeeman splitting of $400\,{\rm \mu eV}$ at $200\,{\rm mT}$ [19, 20, 22, 23]. Both charging energy and Zeeman splitting are larger than the usual induced superconducting gap inside the quantum dot $\Delta \sim 100\,{\rm \mu eV}$ [23, 38–41], justifying the approximation in Eq. (2).

The proximity of the middle dot to the superconductor suppresses its $g$-factor [42].

Thus, differently from the outer dots, we consider a finite Zeeman energy $B$. The Hamiltonian of the middle dot is

$$H_{\text{ABS}} = \sum_{\sigma,\sigma'} [\mu_M(\sigma_0)_{\sigma\sigma'} + B(\sigma_z)_{\sigma\sigma'}] c_{M\sigma}^\dagger c_{M\sigma'} + \Delta c_{M\uparrow}^\dagger c_{M\downarrow}^\dagger + \text{h.c.} \tag{3}$$

where $c_{M\sigma}^\dagger$ is the creation operator of electron on the middle dot with spin $\sigma$, $\Delta$ is the induced superconducting gap, and $\sigma_l$ are the Pauli matrices ($l = \{0, x, y, z\}$) acting on the spin subspace. Both spin-polarised, at most singly occupied, dots in Eq. (2) connect to the middle dot by symmetric tunnel barriers with strength $t$. The barrier $t$ controls both normal and spin-orbit tunneling processes:

$$H_t = t \sum_{i=L,R} \cos\theta_i c_{i\sigma_i}^\dagger c_{M\sigma} + it \sum_{i=L,R} \sum_{\sigma'} (\sigma_y)_{\sigma_i\sigma'} \sin\theta_i c_{i\sigma_i}^\dagger c_{M\sigma'} + \text{h.c.} , \tag{4}$$

where $\theta_i$ is the spin precession angle from dot $i$ to the middle island. Thus, the total Hamiltonian is

$$H = H_d + H_{\text{ABS}} + H_t . \tag{5}$$

We obtain the effective low-energy Hamiltonian in the weak-coupling limit, $t \ll \Delta$, through a Schrieffer–Wolff transformation (the derivation is in Appendix A) [43, 44]:

$$\tilde{H} = \sum_i \epsilon_i^{\sigma_i\sigma_j} c_{i\sigma_i}^\dagger c_{i\sigma_i} + \sum_{i,j} \Gamma_{\sigma_i\sigma_j} c_{i\sigma_i}^\dagger c_{j\sigma_j} + \Lambda_{\sigma_i\sigma_j} c_{i\sigma_i}^\dagger c_{j\sigma_j}^\dagger + \text{h.c.} , \tag{6}$$

where $\epsilon_i^{\sigma_i\sigma_j}$ is the renormalised onsite energy of dot $i$, $\Gamma_{\sigma_i\sigma_j}$ is the ECT rate and $\Lambda_{\sigma_i\sigma_j}$ is the CAR rate. While $t \neq 0$, we do not vary the chemical potential of the outer dots, $\mu_L = \mu_R = 0$. For simplicity, we also assume no Zeeman splitting within the middle dot $B = 0$ and that the spin precession angles are symmetric $\theta_L = \theta_R = \theta$ (see Appendix A for more general form). In such case, the effective parameters for the anti-parallel spin configuration are:

$$\Lambda_{\uparrow\downarrow} = \kappa\Delta\cos(2\theta) , \quad \Gamma_{\uparrow\downarrow} = -i\kappa\mu_M\sin(2\theta) , \tag{7}$$

and for the parallel channel:

$$\Lambda_{\uparrow\uparrow} = -i\kappa\Delta\sin(2\theta) , \quad \Gamma_{\uparrow\uparrow} = -\kappa\mu_M\cos(2\theta) , \tag{8}$$

where

$$\kappa = t^2/(\Delta^2 + \mu_M^2 - B^2) . \tag{9}$$

Both onsite corrections terms are equal:

$$\epsilon_L^{\sigma_i\sigma_j} = \epsilon_R^{\sigma_i\sigma_j} = \kappa\mu_M. \tag{10}$$

We observe that the magnitude of $\Lambda_{\sigma_i\sigma_j}$ is maximum at $\mu_M = 0$ and drops with increasing chemical potential $\mu_M$. On the other hand, $\Gamma_{\sigma_i\sigma_j}$ has maxima at finite $\mu_M$. The magnitude of both processes depends on the spin-precession angle $\theta$ and spin configuration of the outer dots as shown in Fig. 2 (a) and (b). To ensure that operation times for $\mathcal{U}_2$ and $\mathcal{U}_3$ are similar, the convenient regime is where $\max\Gamma_{\sigma_i\sigma_j} \sim \max\Lambda_{\sigma_i\sigma_j}$.

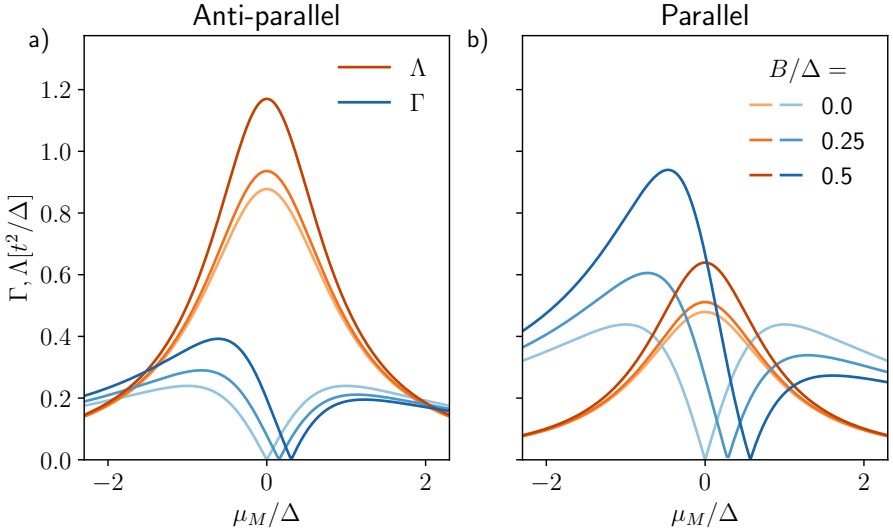

Figure 2: Absolute value of $\Lambda$ (blue) and $\Gamma$ (orange) as a function of $\mu_M$ for different values of $B$ for the anti-parallel configuration (a) and the parallel configuration (b). System parameters are $t = 0.15$, $\theta_L = 0.7$ and $\theta_R = 0.3$ where $\theta_L \neq \theta_R$ for generality.

## 3.2  Capacitive coupling

The electrostatic energy between the two dots is [45]:

$$H_C = \sum_{i=L,R} \upsilon_i c^\dagger_{i\sigma_i} c_{i\sigma_i} + U_m c^\dagger_{L\sigma_L} c_{L\sigma_L} c^\dagger_{R\sigma_R} c_{R\sigma_R} \; , \tag{11}$$

where $U_m = C_m e^2/\tilde{C}$ is the mutual interaction between the two dots,

$$\upsilon_{L/R} = \frac{C_{R/L}(2n_{g,L/R} + 1) + C_m n_{g,R/L}}{2\tilde{C}} \tag{12}$$

is the renormalization to the onsite energy, $\tilde{C} = C_L C_R - C_m^2$, $C_L$ and $C_R$ are the capacitances of the left and right dots, $C_m$ is the mutual capacitance, and $n_{g,i}$ is the charge offset in the site $i$. Notice that we consider single-occupation of the dots in (11). This approximation is valid when $U_m \ll \Delta \ll e^2 C_{L/R}/\tilde{C}$ because the charging energy renormalization due to the mutual capacitance is negligible in this regime. The last term in (11) gives the Coulomb interaction between the dots required to implement $\mathcal{U}_4$.

# 4  Fermionic quantum gates

## 4.1  Unitary gate operations

To achieve the fermionic quantum operations defined in Eq. (1), we require specific time-dependent profiles that vary tunable system parameters. In this case, we control the following system parameters through Eqs. (11) and (6): left and right plunger gates ($\mu_L, \mu_R$), middle plunger gate ($\mu_M$), tunnel gates ($t$, we treat the two tunnel gates together), and mutual capacitance ($C_m$). For simplicity, we only consider square pulses in time

$$H(\tau) = H_P(S)[\Theta(\tau) - \Theta(\tau - \tau_P)] \tag{13}$$

where $\Theta(\tau)$ is the Heaviside step function, $\tau$ is time and $\tau_P$ is the duration of the pulse. We define the pulse Hamiltonian $H_P(S)$ as a constant total Hamiltonian where $S = \{t, \mu_M, ...\}$ are non-zero system parameters in the pulse. For example, $H_P(\{t\})$ is a constant Hamiltonian with all system parameters zero except the tunnel coupling $t$. We set the idle (reference) Hamiltonian to one where all gates are zero, $t = \mu_L = \mu_R = \mu_M = U_m = 0$. Thus, the time-evolution operator simplifies to

$$\mathcal{U}(\tau_2, \tau_1) = \exp\left[-\frac{i}{\hbar}\int_{\tau_1}^{\tau_2} d\tau'\ H(\tau')\right] = \exp\left[-\frac{i}{\hbar}H_P(S)\tau_P\right] . \tag{14}$$

where $\tau_2, \tau_1$ are the initial and final times, and $\tau_P$ is the duration of the pulse. In practice, the transition between the idle Hamiltonian and $H_P$ in Eq. (13) is not instantaneous but ramps up smoothly over a time $\tau_R$ to minimize non-adiabatic transitions.

We engineer the unitary operations as an ordered sequence of pulses defined in Eq. (14). For simplicity, we assume no Zeeman splitting in the middle dot, $B = 0$, and leave the discussion of the more general case to section 4.2. In this case, the minimal pulse sequence scheme which implements the gates in Eq. (1) is:

1. onsite operation:

$$\mathcal{U}_1 = \exp\left[-\frac{i}{\hbar}H_P\left(\{\mu_L, \mu_R\}\right)\tau_P\right] ; \tag{15}$$

2. hopping operation:

$$\begin{aligned}\mathcal{U}_2 &= \exp\left[-\frac{i}{\hbar}H_P\left(\{\mu_L = \mu, \mu_R = \mu\}\right)\tau_P^{(4)}\right] \times \exp\left[-\frac{i}{\hbar}H_P\left(\{t\}\right)\tau_P^{(3)}\right] \\ &\times \exp\left[-\frac{i}{\hbar}H_P\left(\{\mu_L = \mu, \mu_R = \mu\}\right)\tau_P^{(2)}\right] \times \exp\left[-\frac{i}{\hbar}H_P\left(\{t, \mu_M\}\right)\tau_P^{(1)}\right]\end{aligned} \tag{16}$$

3. superconducting pairing operation:

$$\mathcal{U}_3 = \exp\left[-\frac{i}{\hbar}H_P\left(\{t\}\right)\tau_P\right] ; \tag{17}$$

4. Coulomb interaction operation:

$$\mathcal{U}_4 = \exp\left[-\frac{i}{\hbar}H_P\left(\{\mu_L, \mu_R\}\right)\tau_P^{(2)}\right] \times \exp\left[-\frac{i}{\hbar}H_P\left(\{U_m\}\right)\tau_P^{(1)}\right] ; \tag{18}$$

where we indicate as $\tau_P^{(i)}$ the duration of the $i$-th pulse.

In the above scheme, the operations $\mathcal{U}_1$ and $\mathcal{U}_3$ require a single pulse. The gate $\mathcal{U}_1$ requires a single pulse because the dots are uncoupled from one another and the plunger gates affect the onsite energies without inducing any sort of coupling between the dots. Similarly, $\mathcal{U}_3$ is also a single operation because the CAR rate is maximum at $\mu_M = 0$ whereas both ECT rate and onsite corrections are zero according to Eqs. (7— 10). We show the time-dependent simulation of the $\mathcal{U}_3$ gate in Fig. 3.

On the other hand, the first pulse of Eq. (16) introduces finite onsite corrections to the outer dots and CAR according to Eqs. (7— 10). Since the onsite corrections are equal, only a global phase factor is accumulated within the odd fermion parity sector. On the other hand, both onsite corrections and CAR result in undesired rotations within the even fermion parity subspace. We undo these operations with an Euler rotation using two orthogonal operations, resulting in the three subsequent pulses in Eq. (16). We show the time-dependent simulation of the $\mathcal{U}_2$ gate in Fig. 5 (Appendix C). Similarly, the Coulomb operation in Eq. (18) also requires a correction pulse with the plunger gates because the mutual capacitance $C_m$ renormalizes the onsite energies in the outer dots, as shown in Eq. (11).

## 4.2 Finite Zeeman splitting in the middle dot

The presence of Zeeman splitting in the middle dot $B$ introduces an asymmetric onsite renormalisation $\epsilon_L^{\sigma_i\sigma_j} \neq \epsilon_R^{\sigma_i\sigma_j}$ and a shift in the minima of $\Gamma_{\sigma_i\sigma_j}$ shifts away from $\mu_M = 0$ (see Appendix A), as shown in Fig. 2. These changes affect the prescriptions for $\mathcal{U}_2$ and $\mathcal{U}_3$ since these operations require finite $t$.

The asymmetric onsite corrections break the orthogonality between $\mathcal{U}_1$ and the unitary operation prescribed in Eq. (16). It is still possible to implement the $\mathcal{U}_2$ with two non-orthogonal rotation axes in the odd fermion parity sector with additional operations to compensate for the non-orthogonality [46].

The operation in Eq. (17) also introduces finite $\Gamma_{\sigma_i\sigma_j}$ in the odd parity sector. We show in Appendix B that anti-parallel spin configuration with symmetric spin-orbit precession $\theta_L = \theta_R$ removes the shifting $\Lambda$ minima away from $\mu_M = 0$ and restores the orthogonality of the operations within the even parity sector.

## 4.3 Gate performance

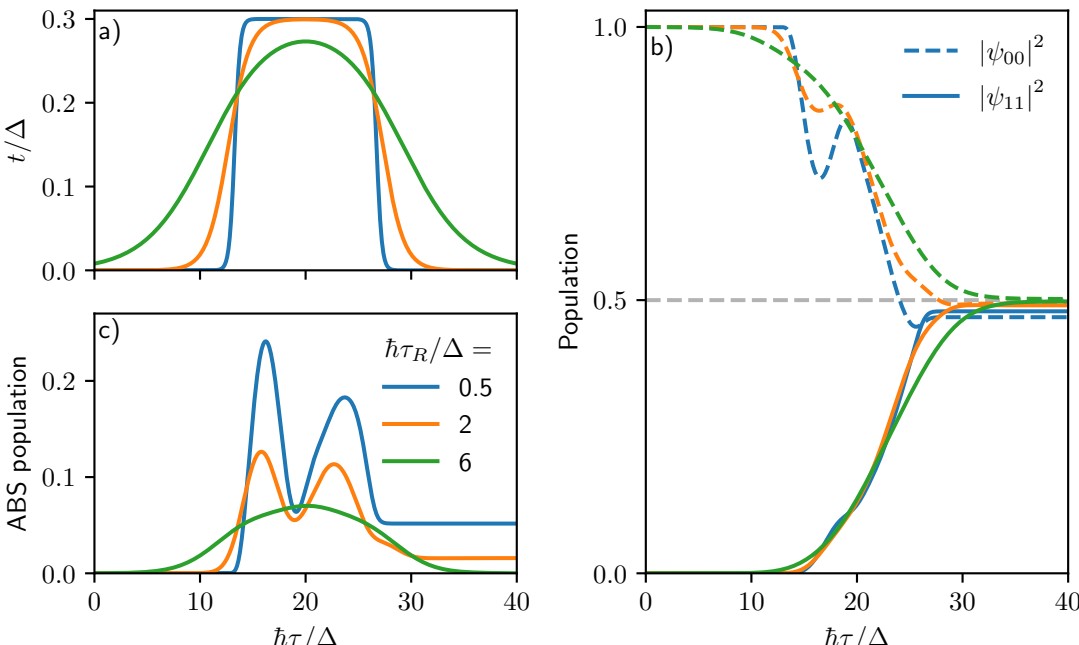

Figure 3: Time-dependent simulation of pairing gate $\mathcal{U}_3$ acting on an initial vacuum state with different pulse rise time $\tau_R$ profiles. The vacuum population is $|\psi_{00}|^2$ whereas the double occupation population (with middle dot unoccupied) is $|\psi_{11}|^2$. Longer pulses (a) result in a smoother transient population profile (b) and less leakage into the middle ABS state (c). The configuration is the spin-antiparallel with finite Zeeman field within the middle dot $B/\Delta = 0.2$ and symmetric spin-orbit precession $\theta_L = \theta_R = \pi/8$

Switching on the pulse in Eq. (13) happens over a finite rise time $\tau_R$. Short rise times $\tau_R$ induce transitions from the LFM dots into the middle ABS at energy $\sim \Delta$ which limits the performance of the gates. To avoid such transitions, the pulse times need to be $\tau_R \gg \hbar/\Delta$. In Fig. 3 we show the time-dependent simulation of the gate $\mathcal{U}_3$ with different rise times. We find that rise times $\tau_R > 2\hbar/\Delta$ ensures negligible transitions into the ABS.

In a system of $\Delta = 100\,\mu\text{eV}$ that corresponds to rise times of $\tau_R > 13\,\text{ps}$.

Current tunable capacitors [35, 36] vary over a limited range. The upper limit for the ratio between the maximum and minimum capacitance $r = C_{\text{off}}/C_{\text{on}}$ is $r \approx 40$ [35, 36]. Thus, there is a non-negligible residual capacitance between the dots when the $\mathcal{U}_4$ gate is off. This residual capacitance acts as an unwanted source of phase and limits the performance of the device. Because such error is coherent, we argue it is possible to offset it after each or a few operations with a compensating $\mathcal{U}_4$ pulse. However, since $[\mathcal{U}_3, \mathcal{U}_4] \neq 0$, the $\mathcal{U}_3$ operation would require similar compensation pulses to Eq. (16) to offset the effect of the residual capacitor.

# 5 Charge neutral local fermionic modes

Although the device in Fig. 1 we proposed has the ingredients to implement universal fermionic gates, further work is required to mitigate the main sources of errors. Because of its similarities to a quantum dot charge qubit, we expect the limiting decoherence mechanism to be the same—charge noise [47]. Typical coherence times are of the order of a few nanoseconds [28, 47–49]. In comparison, Dvir et al. [23] report CAR/ECT strengths that set a lower bound for gate pulse durations of $\hbar/\Gamma_{\text{CAR/ECT}} \approx 50\,\text{ps}$. This minimal device therefore requires gate pulses with sub-nanosecond duration to operate the device within the charge coherence time, posing a significant requirement on control electronics. As an improved alternative, we consider the device shown in Fig. 2 in which all quantum dots are proximitized by a superconductor, so that the local fermionic modes become Andreev quasiparticles. Because Andreev states are linear combinations of electron and hole-like excitations, it is then possible to design a device that operates with charge-neutral fermions. As a consequence, the device becomes quadratically protected against charge noise. A similar idea to avoid charge noise in fermion-parity qubits was recently proposed [50]. Furthermore, a recent work estimated that proximitizing a quantum dot increases the dephasing time to 200 ns [51]. Furthermore, mitigation of the charge noise allows implementation of error-correction codes for fermionic systems [52].

To illustrate the idea, we consider a spinful isolated quantum dot $i$ with Zeeman splitting $B_i$ and onsite charging energy $U_i$. In presence of a superconducting gap $\Delta_i$, the dot hosts Andreev quasiparticles with a Hamiltonian:

$$H_{i,N} = (\epsilon_i + B_i)\,\gamma_{i,\uparrow}^\dagger \gamma_{i,\uparrow} + (\epsilon_i - B_i)\,\gamma_{i,\downarrow}^\dagger \gamma_{i,\downarrow} + U_i \gamma_{i,\uparrow}^\dagger \gamma_{i,\uparrow} \gamma_{i,\downarrow}^\dagger \gamma_{i,\downarrow}\,, \tag{19}$$

where

$$\epsilon_i = \sqrt{\Delta^2 + \delta\mu_i^2} - U_i/2\,, \tag{20}$$

$$\mu_i = \delta\mu_i - U_i/2\,, \tag{21}$$

and $\gamma_{i,\sigma}$ are the annihillation operators of Andreev quasiparticles. When the chemical potential detuning is small $\delta\mu/\Delta_i \ll 1$, the Andreev quasiparticles are equal weight superpositions of electrons and holes:

$$\gamma_{i,\uparrow}^\dagger = u c_{i,\uparrow}^\dagger + v c_{i,\downarrow}\,, \quad \gamma_{i,\downarrow}^\dagger = u c_{i,\downarrow}^\dagger - v c_{i,\uparrow}\,, \tag{22}$$

with the components

$$u = \frac{1}{\sqrt{2}}\left(1 + \frac{\delta\mu_i}{2\Delta_i}\right) + \mathcal{O}(\delta\mu_i^2)\,, \quad v = \frac{1}{\sqrt{2}}\left(1 - \frac{\delta\mu_i}{2\Delta_i}\right) + \mathcal{O}(\delta\mu_i^2)\,. \tag{23}$$

The charge operator corresponds to the Hamiltonian derivative with respect to the chemical potential, and its expectation value for the singly-occupied Andreev quasiparticle states is:

$$\left\langle \gamma_{i,\sigma} \left| \frac{dH_i}{d\delta\mu_i} \right| \gamma_{i,\sigma} \right\rangle = \frac{\delta\mu_i}{\Delta_i} \; , \tag{24}$$

which vanishes when $\delta\mu_i = 0$. Such a chargeless state reduces its coupling to charged sources of noise, however, it is also insensitive to the charge-sensing read-out procedure proposed in Sec. 2. Therefore, during read-out, we propose to first detune the chemical potential $\mu_i$ from the charge-neutral sweetspot to manifest charge. Another way to control the charge of an ABS is via flux-tuning [53–58], but we do not consider it here because it requires additional superconductors and introduces sensitivity to flux noise.

In order to use the charge-neutral fermions as local fermionic modes, we require the vacuum state and a single Andreev quasiparticle state to be the lowest energy states, with all the rest of the states removed away far in energy. We achieve this for dot $i$ that stores LFM by fulfilling the inequality $|U_i/2 + B_i - \Delta_i| \ll B_i$. This condition is satisfied when the charging energy is sufficiently small, and the superconducting gap is comparable to the Zeeman splitting. We suggest the device depicted in Fig. 4 to control these parameters. Differently from the device in Fig. 1, we add a tunable gate-controlled coupling $t_{\Delta,i}$ between the dots and the superconductor that controls the proximity gap $\Delta_i$, the $g$-factor renormalization, and the screening of the Coulomb potential $U_i$. We refer to the dots $L$ and $R$ that store LFMs in Fig. 1 as the outer dots whereas the middle dot $M$ mediates interaction between them.

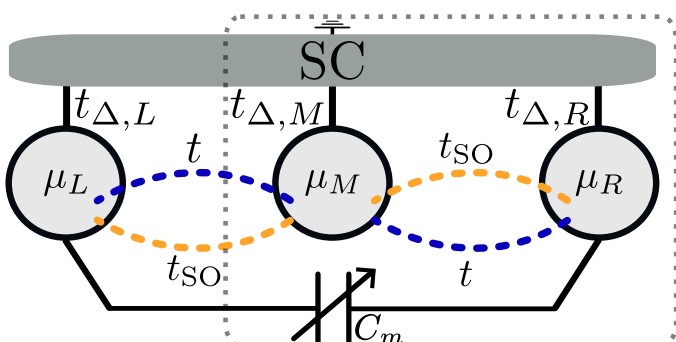

Figure 4: Alternative layout for a device with charge-neutral local fermionic modes. Differently from the device depicted in Fig. 1, we consider tunneling between the quantum dots and the superconductor, $t_{\Delta,i}$. Controlling these additional barriers allows tuning the induced superconducting gap.

## 5.1    Gates

Unlike the charged fermions, which implement the onsite gate using a plunger gate pulse (15), the charge-neutral fermions require fixing the plunger gates at the sweet spot. Instead, we utilize the tunnel barrier $t_{\Delta i}$ connecting the outer dot to the superconducting island. In the weak and moderate coupling regime, tunnel barrier controls the induced superconducting gap $\Delta_i \sim t_{\Delta i}^2/\Delta$, and the energy of the Andreev quasiparticles in Eq. (19). This, therefore yields the desired onsite operation in the neutral fermion device:

$$\mathcal{U}_1 = \exp\left[ -\frac{i}{\hbar} H_P\left(\{t_{\Delta_L}, t_{\Delta_R}\}\right) \tau_P \right] \; . \tag{25}$$

Similarly to the charged fermion case, we couple two neutral fermions through a middle dot strongly coupled to a superconductor, as shown in Fig. 4. We choose a symmetric set of

parameters for the outer neutral fermions: $\Delta_L = \Delta_R = \Delta_N$, $\theta_L = \theta_R = \theta$ and $B_L = B_R = B_N$. Furthermore, we neglect Zeeman splitting in the middle dot because leading-order effects are $\sim t^2 B^2/\Delta^4$. Again, we assume a weak-coupling limit, $t \ll \Delta, \Delta_N, B_N$, and perform a Schrieffer–Wolff transformation to obtain an effective low-energy model [43]. From the effective model, we obtain the coupling strengths between neutral fermions:

$$\Lambda_N = i\kappa\mu_M \sin(2\theta) , \quad \Gamma_N = \kappa(\Delta + \Delta_N - B_N) \cos(2\theta) , \tag{26}$$

where

$$\kappa = t^2 / \left[ \Delta^2 + \mu_M^2 - (\Delta_N - B_N)^2 \right] \tag{27}$$

and a renormalized onsite energy:

$$\tilde{\epsilon}_N = (\Delta_N - B_N) + \kappa(\Delta + B_N - \Delta_N) . \tag{28}$$

Compared to the charged fermion case in Eq. (8), we observe that the terms that implement $\mathcal{U}_2$ and $\mathcal{U}_3$ in Eq. (26) behave in a qualitatively opposite way: $\Lambda_N = 0$ and $\Gamma_N$ is maximal at $\mu_M = 0$.

Because of the swapped nature of $\mathcal{U}_2$ and $\mathcal{U}_3$, the hopping operation in the neutral fermion basis requires fewer steps:

$$\mathcal{U}_2 = \exp\left[ -\frac{i}{\hbar} H_P \left( \{ t_{\Delta_L} = t_{\Delta_N}, t_{\Delta_R} = t_{\Delta_N} \} \right) \tau_P^{(2)} \right] \times \exp\left[ -\frac{i}{\hbar} H_P \left( \{t\} \right) \tau_P^{(1)} \right] . \tag{29}$$

Likewise, by tuning $\Delta_N \approx B_N$ such that there is no onsite energy in Eq. (28), pairing operation is simple in the even parity subspace and only requires a single $\mathcal{U}_2$ correction pulse in the odd parity subspace:

$$\mathcal{U}_3 = \exp\left[ -\frac{i}{\hbar} H_P \left( \{ t_{\Delta_L} = t_{\Delta_N}, t_{\Delta_R} = t_{\Delta_N} \} \right) \tau_P^{(3)} \right] \times \exp\left[ -\frac{i}{\hbar} H_P \left( \{t\} \right) \tau_P^{(2)} \right]$$
$$\times \exp\left[ -\frac{i}{\hbar} H_P \left( \{t, \mu_M\} \right) \tau_P^{(1)} \right] . \tag{30}$$

To confirm the validity of pairing and hopping operations, we perform their time-dependent simulation in Fig. 7 (Appendix C) where we consider the effect of a constant full model Hamiltonian in Eq. (19). In a system with the middle strongly-coupled dot proximitized to $\Delta = 100\,\mu\text{eV}$, we predict the gate operation duration down to 1 ns.

We neglect other spin channels and doubly occupied states in the charged fermions regime because $B_i, U_i \gg \Delta$ is a reasonable approximation. With neutral fermions, this approximation is not valid, so we systematically eliminate the remaining states in the outer dots using a Schrieffer–Wolff perturbative expansion in $U_m/\Delta_N$ [43]:

$$H_C = \sum_{i=L,R} \frac{3U_m^2}{\Delta_N} \gamma_{i\downarrow}^\dagger \gamma_{i\downarrow} - \frac{U_m^2}{\Delta_N} \gamma_{L\downarrow}^\dagger \gamma_{L\downarrow} \gamma_{R\downarrow}^\dagger \gamma_{R\downarrow} , \tag{31}$$

Thus, we can still implement $\mathcal{U}_4$ through the mutual capacitance $C_m$ between the outer dots even in the charge-neutral regime:

$$\mathcal{U}_4 = \exp\left[ -\frac{i}{\hbar} H_P \left( \{ t_{\Delta_L} = t_{\Delta_N}, t_{\Delta_R} = t_{\Delta_N} \} \right) \tau_P^{(2)} \right] \times \exp\left[ -\frac{i}{\hbar} H_P \left( \{U_m\} \right) \tau_P^{(1)} \right] . \tag{32}$$

Finally, because now the effective interaction is quadratic in $U_m^2$, the device becomes less sensitive to the decoherence effects of residual capacitance discussed in Sec. 4.3.

## 5.2    Comparison with charged fermions

Opposite to charged fermions, neutral fermions are quadratically protected against plunger gates' charge noise. In an isolated ($t = 0$) neutral fermion mode, we identify the noise in superconductor tunnel gates $t_{\Delta i}$ as another source of decoherence, because it modulates the induced superconducting gap $\Delta_i$. However, the tunnelling rate is usually less sensitive than the chemical potential to variations of gate voltage, $\partial \mu_i / \partial V \gg \partial t / \partial V$ [59] and thus the tunnel gate noise contributes less to the overall decoherence. As a result, we expect the neutral fermion devices to have longer coherence times than charged fermion devices.

An exception to the insensitivity to charge noise is the pairing operation. The parameter $\Lambda_N$ in Eq. (26) is linear in $\mu_M$ and, therefore, susceptible to first-order charge noise. However, the coupling prefactor $t^2/\Delta^2$ is much smaller than the main sources of charge noise in the charged fermion regime.

In order to work only with the lowest energy states, we require the charging energy $U_i$ to be much smaller than both Zeeman $B_i$ and the superconducting gap $\Delta_i$. In addition, because the outer dots are weakly-coupled, the induced superconducting gap in the outer dots is smaller than the one in the middle $\Delta_i < \Delta$, which reduces the energy gap of the computation states.

# 6    Future directions

Our proposed device consists of a chain of single-orbital fermionic sites. The device layout is a limiting factor, as it only allows nearest-neighbor in hoppings, superconducting pairing, and electrostatic interactions. The layout limitations are detrimental to effective scalability. Thus, future works could, for example, generalize the model to two-dimensional lattices.

We showed that the proposed device is a minimal example of a fermionic quantum computer. However, we must also emphasize that the high control of the system parameters allows using the same device as a quantum simulator. For example, a chain-like device with the unit cell shown in Fig. 1 at finite $\Gamma_{\sigma_i, \sigma_j}$ and $U_m$ can be directly mapped to the Heisenberg model. Thus, with the superconducting correlations, these devices would be an extension of other quantum dot platforms [60].

We mentioned in Sec. 4 that all tunable capacitors proposed present a residual mutual capacitance $C_{\text{off}}$. The external capacitor is necessary because charge screening in the superconducting island suppresses interdot interactions. On the other hand, a floating superconducting island offers a direct interdot capacitance [61]. In a device with a switch between a floating and grounded superconductor, there would be direct control of the mutual capacitance [62]. Moreover, the charge screening due to the grounded superconducting island sets $C_{\text{off}} \to \infty$, removing the need to fix offset phases due to the residual capacitance. The complexity of this setup requires further experimental investigation. Thus, we referred to the alternative methods despite their limitations.

# 7    Summary

We showed that Copper pair-splitting devices with tunable capacitors allow a minimal implementation of a fermionic quantum computer. We derived the low-energy Hamiltonian and showed how to implement a universal set of gate operations by tuning experimentally controllable parameters. Moreover, we show how to suppress decoherence due to charge

noise with an alternative device layout where all quantum dots have independent tunable couplings to the superconducting reservoir. We achieve the insensitivity to charge noise operating in a regime where the local fermionic modes are charge-neutral. Based on the low-energy theory, we also studied optimal regimes for the operation of both devices. By repeating the unit cell, it is possible to use the system as a static simulator of one-dimensional fermionic chains.

# Acknowledgements

The authors acknowledge the inputs of: Isidora Araya Day on perturbation theory calculations; Mert Bozkurt and Chun-Xiao Liu on the device conception and development of an effective model; Christian Prosko, Valla Fatemi, David van Driel, Francesco Zatelli, and Greg Mazur on the experimental feasibility.

### Data Availability

All code used in the manuscript is available on Zenodo [43].

### Author contributions

A.A. and M.W. formulated the initial project idea and advised on various technical aspects. K.V. and A.M. supervised the project. K.V., A.M. and J.T. developed the effective model and the device design. K.V., A.M., J.T., and S.M. constructed the gate operations. K.V. performed the time-dependent calculations. All authors contributed to the final version of the manuscript.

### Funding information

The project received funding from the European Research Council (ERC) under the European Union's Horizon 2020 research and innovation program grant agreement No. 828948 (AndQC). The work acknowledges NWO HOT-NANO grant (OCENW.GROOT.2019.004) and VIDI Grant (016.Vidi.189.180) for the research funding.

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

## A   Schrieffer–Wolff transformation

We perform a Schrieffer–Wolff transformation to obtain the effective Hamiltonian from Eq. (6). We, first, diagonalize the Hamiltonian of the middle dot in Eq. (3):

$$H_{\text{ABS}} = (\epsilon_{\text{ABS}} + B)\gamma_\uparrow^\dagger \gamma_\uparrow + (\epsilon_{\text{ABS}} - B)\gamma_\downarrow^\dagger \gamma_\downarrow \, , \tag{33}$$

where $\epsilon_{\text{ABS}} = \sqrt{\Delta^2 + \mu_M^2}$, $\gamma_\sigma$ are the annihillation operators of Andreev quasiparticles

$$\gamma_\uparrow^\dagger = u c_{M\uparrow}^\dagger + v c_{M\downarrow} \, , \quad \gamma_\downarrow^\dagger = u c_{M\downarrow}^\dagger - v c_{M\uparrow} \, , \tag{34}$$

and $u$ and $v$ are the coherence factors.

We now define the occupation basis for the many-body states as $|n_L, n_M, n_R\rangle$, where $n_i$ corresponds to the occupation number at the site $i$. Notice that for the middle dot, we define the number operator as $\hat{n}_{M\sigma} = \gamma_{M\sigma}^\dagger \gamma_{M\sigma}$, whereas in the outer dots $\hat{n}_{i\sigma_i} = c_{i\sigma_i}^\dagger c_{i\sigma_i}$. Because we consider $\mu_{L/R}, B \ll \Delta$, in the absence of hopping between the dots,

$$\langle n_L, 0, n_R | H | n_L, 0, n_R \rangle \ll \langle n_L, n_M, n_R | H | n_L, n_M, n_R \rangle \, , \tag{35}$$

for $n_{L/R} \in \{0, 1\}$, and $n_M > 0$. Thus, the states with zero occupation in the middle dot form our low-energy manifold.

An energy $\sim \Delta$ separates the occupied states in the middle dot from the low-energy manifold. In the weak coupling limit $t \ll \Delta$, the high-energy subspace only contributes to the low-energy dynamics through virtual processes. Therefore, we use a Schrieffer–Wolff

transformation to obtain the effective Hamiltonian in the low-energy subspace in Eq. (6). Whenever $\mu_L = \mu_R = 0$, the terms in Eq. (6) for the anti-parallel spin configuration are:

$$\epsilon_R^{\uparrow\downarrow} = \kappa \left( -2B \sin^2 \left( \theta_R \right) + B + \mu_M \right), \quad \epsilon_L^{\uparrow\downarrow} = \kappa \left( -2B \cos^2 \left( \theta_L \right) + B + \mu_M \right), \tag{36}$$

$$\Lambda_{\uparrow\downarrow} = \kappa\Delta \cos(\theta_L + \theta_R) , \quad \Gamma_{\uparrow\downarrow} = -i\kappa \left[ \mu_M \cos \left( \theta_L + \theta_R \right) - B \sin \left( \theta_L - \theta_R \right) \right] , \tag{37}$$

and for the parallel configuration:

$$\epsilon_R^{\uparrow\uparrow} = \kappa \left( -2B \cos^2 \left( \theta_R \right) + B + \mu_M \right), \quad \epsilon_L^{\uparrow\uparrow} = \kappa \left( -2B \cos^2 \left( \theta_L \right) + B + \mu_M \right), \tag{38}$$

$$\Lambda_{\uparrow\uparrow} = -i\kappa\Delta \sin \left( \theta_L + \theta_R \right) , \quad \Gamma_{\uparrow\uparrow} = -\kappa \left[ \mu_M \cos \left( \theta_L + \theta_R \right) - B \cos \left( \theta_L - \theta_R \right) \right] , \tag{39}$$

where

$$\kappa = t^2/(\Delta^2 + \mu_M^2 - B^2) . \tag{40}$$

At finite $B$, the chemical potential $\mu_M$ at which $\Gamma_{\sigma_i \sigma_j} = 0$ shifts to:

$$\mu_{\text{shift}}^{\uparrow\downarrow} = \frac{B \sin \left( \theta_L - \theta_R \right)}{\sin \left( \theta_L + \theta_R \right)}, \quad \mu_{\text{shift}}^{\uparrow\uparrow} = \frac{B \cos \left( \theta_L - \theta_R \right)}{\cos \left( \theta_L + \theta_R \right)} , \tag{41}$$

for anti-parallel and parallel spin configurations.

## B    Convenience of the anti-parallel spin configuration

### B.1    Orthogonality with symmetric spin precession

In Eq. (36), for the anti-parallel spin configuration, we notice that when the spin precession angles are equal $\theta_L = \theta_R = \theta$, the double occupation onsite energy $\epsilon_L + \epsilon_R = 0$ is zero at $\mu_M = 0$ and the ETC minima shifts disappear as shown in Eq. (41). That restores the orthogonality of operations within the even parity sector, and thus we express $\mathcal{U}_3(\gamma)$ operation as:

$$\mathcal{U}_3(\gamma) = \exp \left[ -\frac{i}{\hbar} H_P \left( \{\mu_{L/R}\} \right) \tau_P^{(2)} \right] \times \exp \left[ -\frac{i}{\hbar} H_P \left( \{t\} \right) \tau_P^{(1)} \right] , \tag{42}$$

where we compensate a finite $\epsilon_L - \epsilon_R$ with an onsite pulse. On the other hand, the hopping operation requires additional operations to compensate for non-orthogonality [46]:

$$\begin{aligned}
\mathcal{U}_2(\beta) = &\exp \left[ -\frac{i}{\hbar} H_P \left( \{\mu_L = \mu, \mu_R = \mu\} \right) \tau_P^{(N+4)} \right] \times \exp \left[ -\frac{i}{\hbar} H_P \left( \{t\} \right) \tau_P^{(N+3)} \right] \\
&\times \exp \left[ -\frac{i}{\hbar} H_P \left( \{\mu_L = \mu, \mu_R = \mu\} \right) \tau_P^{(N+2)} \right] \times \exp \left[ -\frac{i}{\hbar} H_P \left( \{\mu_{L/R}\} \right) \tau_P^{(N+1)} \right] \\
&\times \prod_{j=1}^{N/2} \exp \left[ -\frac{i}{\hbar} H_P \left( \{t, \mu_M\} \right) \tau_P^{(2j)} \right] \times \exp \left[ -\frac{i}{\hbar} H_P \left( \{\mu_{L/R}\} \right) \tau_P^{(2j-1)} \right]
\end{aligned} \tag{43}$$

where $N$ is the number of pulses required to correct for the non-orthogonality within the odd parity sector.

## B.2   Stability and number of operations

To quantify the degree of linear dependence of the operations, we define the following metric

$$\mathcal{L}_o = \sqrt{\frac{(\epsilon_L - \epsilon_R)^2}{(\epsilon_L - \epsilon_R)^2 + \Gamma^2}}, \tag{44}$$

$$\mathcal{L}_e = \sqrt{\frac{(\epsilon_L + \epsilon_R)^2}{(\epsilon_L + \epsilon_R)^2 + \Lambda^2}} \tag{45}$$

for $\mathcal{L}_e$ even $\mathcal{L}_o$ and odd fermion parity sectors. If $\mathcal{L}_{e/o} = 0$, the operations are orthogonal, and the scheme outlined in Section 4 is valid. On the other hand, if $\mathcal{L}_{e/o} = 1$, it is impossible to generate a universal set of operations. To understand how robust the scheme in Eq. (42) is, we consider small deviations from the perfect spin precession case: $\theta_L = \theta$ and $\theta_R = \theta + \delta$. In this case, the metric $\mathcal{L}_{e/o}$ reads:

$$\mathcal{L}_o = \left[ 1 + \left( \frac{\mu_M}{B} \tan 2\theta \right)^2 \right]^{-1/2} + O(\delta) \, , \tag{46}$$

$$\mathcal{L}_e = \delta \left( \frac{B}{\Delta} \right) \tan 2\theta + O(\delta^2) \, . \tag{47}$$

Depending on the linear dependence $\mathcal{L}_{e/o}$ of the hopping and pairing operations, we can estimate the maximal number of pulses required to implement an arbitrary operation [46] within a given fermion parity subspace:

$$\mathcal{N}(\mathcal{L}_{e/o}) = \lceil \frac{\pi}{\arccos\left(\mathcal{L}_{e/o}\right)} \rceil + 1. \tag{48}$$

## C   Time-dependent gate operations

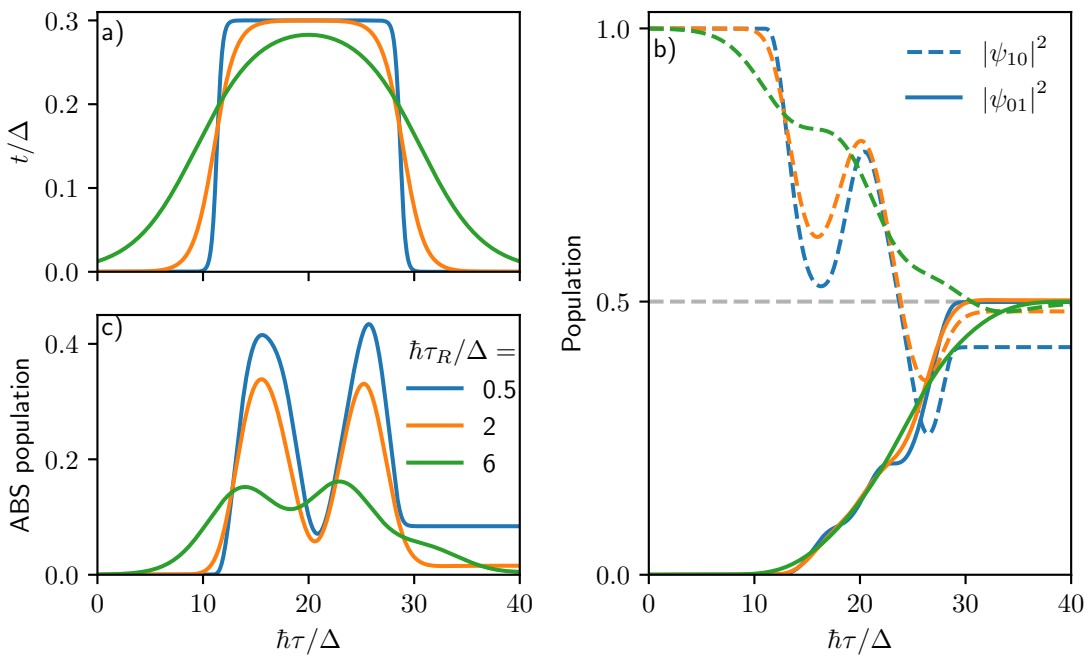

Figure 5: Time-dependent simulation of hopping gate $\mathcal{U}_2$ acting on an initial left singly-occupied state with different pulse rise time $\tau_R$ profiles. The left state population is $|\psi_{10}|^2$ whereas the right state population is $|\psi_{01}|^2$. Longer pulses (a) result in a smoother transient population profile (b) and less leakage into the middle ABS state (c). The configuration is the spin-antiparallel with finite Zeeman field within the middle dot $B/\Delta = 0.2$ and symmetric spin-orbit precession $\theta_L = \theta_R = \pi/8$

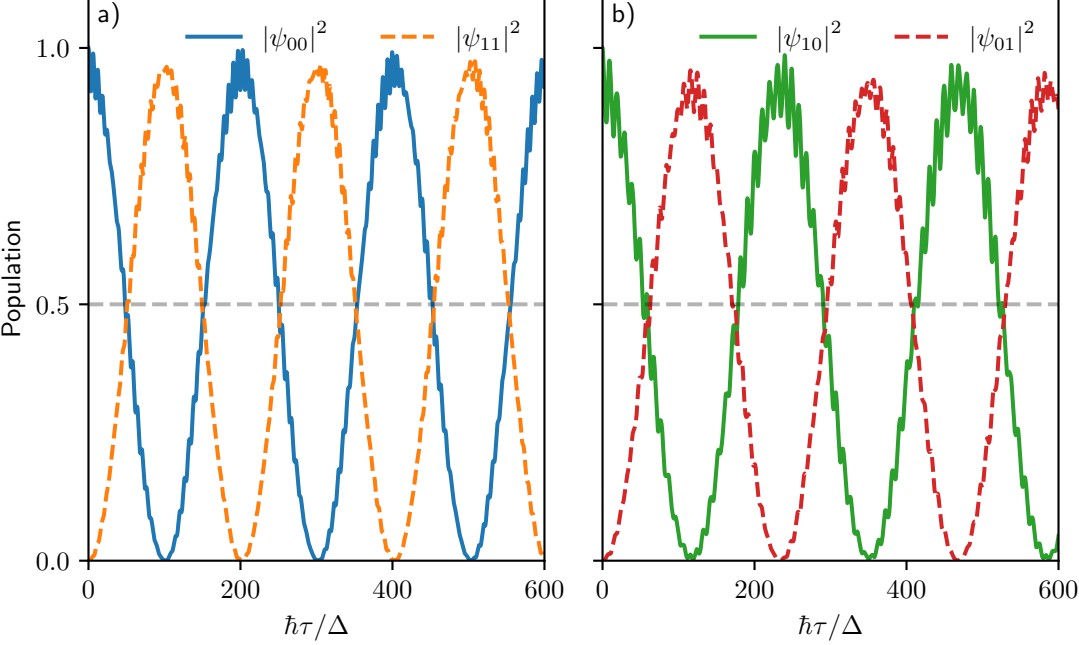

Figure 6: The time-dependent evolution due to a constant charged fermion Hamiltonian showing (a) $\mathcal{U}_3$ pairing and (b) $\mathcal{U}_2$ hopping operations. The parameters are $t/\Delta = 0.15, \alpha = \pi/8$ with $\mu_M/\Delta = 0$ for (a) and $\mu_M/\Delta = 0.45$ for (b).

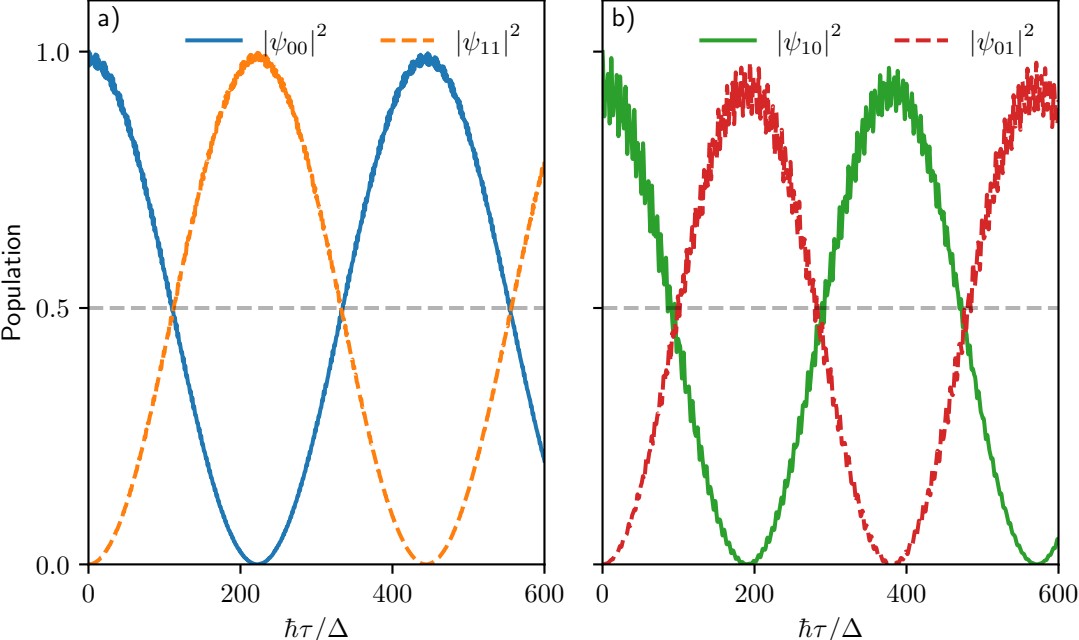

Figure 7: The time-dependent evolution due to a constant neutral fermion Hamiltonian showing (a) $\mathcal{U}_3$ pairing and (b) $\mathcal{U}_2$ hopping operations. The parameters are $t/\Delta = 0.15, B_N/\Delta = 0.3, \Delta_N/\Delta = 0.297, \alpha = \pi/3.2$ with $\mu_M/\Delta = 2.5$ for (a) and $\mu_M/\Delta = 0$ for (b).