# Peer review of "Fermionic quantum computation with Cooper pair splitters"

_SciPost Physics_

## Round 1 · Referee Report · Anonymous (Referee 1) · 2023-10-16

Strengths

1- Establishes a clear simplified model, with a hierarchy of scales that seems reasonable and is experimentally relevant 2- Succinct: to the point, important issues are discussed

Weaknesses

1- The introduction should present the start-of-the-art on physical implementations of LFM bases devices and proposals thereof, if any. 2- Zeeman splitting in the central dot could potentially hamper actual experimental implementations and make it complicated.

Report

This work is a theoretical study of a basic unit cell of a quantum
computing devices using local fermionic modes. The unit cell consists
of two spin-polarized quantum dots and a middle superconducting island
(or actually a proximitized non-interacting quantum dot). A
non-interacting Hamiltonian is proposed, and then further simplified
by removing all states with a non-zero number of Andreev
quasiparticles in the middle dot using a Schrieffer-Wolff
transformation. The authors then show that the universal set of gate
operators can be implemented using appropriate pulsing sequeces and
discuss various implementation aspects.

The proposed microscopic model seems appropriate for a first proof of
concept theoretical work, the approximation involved are reasonable
and realistic, the derivations appear correct, and I don't see any
a-priori reasons why this scheme could not some day be physically
implemented, although it would certainly be very challenging. There is
sufficient attention to various experimentally relevant issues and
non-idealities that could affect device operation. The work is also
timely, since devices with a similar structure are currently being
intensively investigated. I thus recommend this work for publication,
because it opens a new research direction with potential for follow-up
work (e.g. actual implementation) and it meets the general acceptance
criteria (clarity, structure, level of detail, reference to existing work,
resources, clear conclusion).

Requested changes

I have only minor comments and suggestions:

1- From the introduction and Fig. 1 it is not immediately clear how the "unit cell" would be stacked in a large-scale device. In fact, the expression "unit cell" itself is perhaps inappropriate, since one would not stack together QD_L-QD_M-QD_R units as QD_L1-QD_M-QD_R1-QD_L2-QD_M-QD_R2-..., but presumably form a chain such as QD_1-QD_M-QD_2-QD_M-QD_3-...

2- "Notice that we consider singly-occupied dots in (11)." Probably you mean "at most singly occupied"?

3- Starting on p. 6, there appears a new parameter U (just before Eq. 14, as an argument in Eq. 18). I suppose this should really be U_m from Eq. 11.

4- .. at finite Gamma and U .. Yet another quantity appears here, Gamma. It is never defined.

5- Are there alternative proposals for computing devices with LFM architecture? What are the advantages of the scheme proposed in this work? Any disadvantages?

There are also some typos:

6- Typo in text after Eq. (1): U_1(delta) should be U_4(delta)

7- missing f in Schrieffer-Wolff on p. 4

8- puls -> pulse

9- "an asymmetric the onsite" -> "an asymmetric onsite"

10- In Fig. 2 -> In Fig. 3 (first paragraph of Sec. 4.3)

  • validity: high
  • significance: good
  • originality: good
  • clarity: high
  • formatting: perfect
  • grammar: perfect

Author:  Kostas Vilkelis  on 2024-04-09  [id 4404]

(in reply to Report 1 on 2023-10-16)

We thank the referee for their report and for considering our work experimentally relevant.

The introduction should present the start-of-the-art on physical implementations of LFM bases devices and proposals thereof, if any.

We thank the referee for their suggestion. We have added the corresponding paragraph to the introduction.

Zeeman splitting in the central dot could potentially hamper actual experimental implementations and make it complicated.

We agree that the Zeeman splitting in the central dot makes the experimental implementation more complicated.

1- From the introduction and Fig. 1 it is not immediately clear how the "unit cell" would be stacked in a large-scale device. In fact, the expression "unit cell" itself is perhaps inappropriate, since one would not stack together QD_L-QD_M-QD_R units as QD_L1-QD_M-QD_R1-QD_L2-QD_M-QD_R2-..., but presumably form a chain such as QD_1-QD_M-QD_2-QD_M-QD_3-...

The referee is correct. We have updated the device schematic to make this point clearer.

2- "Notice that we consider singly-occupied dots in (11)." Probably you mean "at most singly occupied"? 3- Starting on p. 6, there appears a new parameter U (just before Eq. 14, as an argument in Eq. 18). I suppose this should really be U_m from Eq. 11. 4- .. at finite Gamma and U .. Yet another quantity appears here, Gamma. It is never defined.

We have addressed these remarks in the updated version of the manuscript.

5- Are there alternative proposals for computing devices with LFM architecture? What are the advantages of the scheme proposed in this work? Any disadvantages?

We have added a discussion of alternative platforms to the introduction.

There are also some typos: 6- Typo in text after Eq. (1): U_1(delta) should be U_4(delta) 7- missing f in Schrieffer-Wolff on p. 4 8- puls -> pulse 9- "an asymmetric the onsite" -> "an asymmetric onsite" 10- In Fig. 2 -> In Fig. 3 (first paragraph of Sec. 4.3)

We fixed these typos in the updated manuscript.

Attachment:

diff_manuscript_YAM4v2x.pdf

---

## Round 1 · Referee Report · Anonymous (Referee 2) · 2023-11-7

Strengths

Timely and relevant for near-term experiments.
Well written and for the most part easy to understand.

Weaknesses

The expected short coherence times seems to severely limit the usefulness for actual quantum computing.
The results shown are rather limited, with only simulations of one particular gate being investigated in detail.

Report

The manuscript contains a proposal for, and theoretically analyzes of, a way to implement quantum computing based on local fermionic modes. The proposal is based on a Cooper pair splitter geometry, where two spin-polarized quantum dots are coupled by cotunneling and crossed Andreev reflection via a narrow superconducting region. The manuscript is well written and the general idea is nice and timely; with several experimental groups already studying similar geometries for other purposes this work has the potential to inspire near-future experiments. In that sense, it can be said to open a pathway for new research. It might thus be suitable for publication in SciPost.

My main concern with this paper is that it seems, to me, that the proposed system has serious problems which makes it unlikely to be successful as a quantum computing platform (and might even prevent a proof-of-principle demonstration). As pointed out be the authors, the setup is susceptible to charge noise, which is known to limit coherence times very severely. It seems that one of the main reasons for using a superconducting system would be to avoid such problems, and indeed the authors mention this possibility, but they do not explore it in this work. A further problem is that the tunable mutual capacitance seems difficult from an experimental perspective. Also in this case, the authors mention an alternative platform to avoid this problem, but do not investigate it further. This begs the question of why the authors choose to investigate this platform, and not a more promising alternative?

In my opinion, it would strengthen the manuscript if these alternative platforms were discussed more thoroughly, including not only their advantages but also disadvantages (which presumably also exist). I also think that the authors should do a more careful analyzes of the performance of the additional gates, now they only show results for U_3.

I also have a few more detailed/minor/technical questions and comments:
1) I’m unsure what is meant with the proposed system being a unit cell of a fermionic quantum computer. What will the coupling to the next cell be, via an additional superconductor or via direct tunneling to another quantum dot?
2) After Eq. (1) it is stated that in the presence of an ancilla, U_4 entangles the even and odd subspaces. I find this statement unclear. What is meant, I presume, is that starting from a superposition of even and odd parities, U_4 leads to an entangled state.
3) The title (Tunnel coupling) of section 3.1 seems strange, this section introduces the whole Hamiltonian (except the capacitive coupling).
4) Above Eq. (7), it is stated that theta_L=theta_R, but this does not seem to be the case in Fig. 2 (according to the caption).
5) Based on experimentally measured amplitudes for CAR and ECT, the authors suggest pulse durations of around 50 ps. To me, this seems beyond the reach of standard electronics, at least if the pulses should be accurately timed to give high fidelity gates. I guess one can use smaller interactions strengths, but then the short expected coherence times become even more problematic …
6) The authors state that Eq. (2) is valid if both charging energy and Zeeman splitting are large. But if Zeeman is large, I don’t see why it would be necessary to assume also a large charging energy. One additional assumption not mentioned is that the energy separation of single-particle orbitals is large.

Requested changes

See report.

  • validity: high
  • significance: ok
  • originality: high
  • clarity: high
  • formatting: excellent
  • grammar: excellent

Author:  Kostas Vilkelis  on 2024-04-09  [id 4403]

(in reply to Report 2 on 2023-11-07)

We thank the referee for considering our work timely and relevant. We agree with both referee's observations about weaknesses of the previous version of our manuscript, which we have addressed in the resubmission.

My main concern with this paper is that it seems, to me, that the proposed system has serious problems which makes it unlikely to be successful as a quantum computing platform (and might even prevent a proof-of-principle demonstration). As pointed out be the authors, the setup is susceptible to charge noise, which is known to limit coherence times very severely. It seems that one of the main reasons for using a superconducting system would be to avoid such problems, and indeed the authors mention this possibility, but they do not explore it in this work.

We thank the referee for this remark, and we agree with the overall concern. Motivated by the referee's query, we have added an extended analysis (section 5) investigating a device with charge neutral local fermionic modes. We describe how to make the device insensitive to charge noise, at a cost of additional complexity of the device and its tuning. We refer to the manuscript for the full discussion.

A further problem is that the tunable mutual capacitance seems difficult from an experimental perspective. Also in this case, the authors mention an alternative platform to avoid this problem, but do not investigate it further. This begs the question of why the authors choose to investigate this platform, and not a more promising alternative?

While a tunable effective capacitance adds complexity to the device, several proposals that we list have been demonstrated experimentally, and are based on relatively well-studied physical phenomena, in particular Refs. 34 (double quantum dot) and 33 (superconducting islands with tunable Josephson junctions). Both implementations were demonstrated to tune the capacitive coupling over an exponentially broad range of energies. The analysis of which implementation of mutual capacitance is most relevant to implement in an experimental device depends on a specific implementation and platform, and therefore we believe that it is outside of the scope of our work.

In my opinion, it would strengthen the manuscript if these alternative platforms were discussed more thoroughly, including not only their advantages but also disadvantages (which presumably also exist). I also think that the authors should do a more careful analyzes of the performance of the additional gates, now they only show results for U_3.

We now compare charged and chargeless local fermionic mode ideas in the new "5.2 Comparison with charged fermions" subsection. We also added Appendix C showing additional time-dependent simulations of the other gates. However, we do not carry out a more comprehensive tunable capacitor comparison because it is beyond the scope of this work.

I also have a few more detailed/minor/technical questions and comments:

We address all these points in the new manuscript version and answer below for completeness.

1) I’m unsure what is meant with the proposed system being a unit cell of a fermionic quantum computer. What will the coupling to the next cell be, via an additional superconductor or via direct tunneling to another quantum dot?

We have updated the device schematic to clearly indicate the unit cell and the additional coupling needs to be through another proximitized mode.

2) After Eq. (1) it is stated that in the presence of an ancilla, U_4 entangles the even and odd subspaces. I find this statement unclear. What is meant, I presume, is that starting from a superposition of even and odd parities, U_4 leads to an entangled state.

We have clarified the statement. In particular, we explained that $U_4$ indeed generates multifermion entanglement starting from a state that is a superposition of local fermion parity.

3) The title (Tunnel coupling) of section 3.1 seems strange, this section introduces the whole Hamiltonian (except the capacitive coupling). 4) Above Eq. (7), it is stated that theta_L=theta_R, but this does not seem to be the case in Fig. 2 (according to the caption).

We have fixed these issues in the updated manuscript.

6) The authors state that Eq. (2) is valid if both charging energy and Zeeman splitting are large. But if Zeeman is large, I don’t see why it would be necessary to assume also a large charging energy. One additional assumption not mentioned is that the energy separation of single-particle orbitals is large.

A large charging energy is necessary to separate a doubly occupied state from an unoccupied one. We now state it in an additional remark in the manuscript. Alternatively, this role is fulfilled by pairing in the newly added neutral fermion implementation.

5) Based on experimentally measured amplitudes for CAR and ECT, the authors suggest pulse durations of around 50 ps. To me, this seems beyond the reach of standard electronics, at least if the pulses should be accurately timed to give high fidelity gates. I guess one can use smaller interactions strengths, but then the short expected coherence times become even more problematic …

Indeed, the original device proposal puts stringent constraints on the gate duration due to charge noise. We now demonstrate that the neutral fermion implementation relaxes this requirement, and we believe that it shows sufficient promise to be attempted experimentally. We keep the charged fermion scheme in the manuscript due to it being simpler, useful as a proof-of-concept, and for completeness.

Attachment:

diff_manuscript.pdf

---

## Round 1 · Referee Report · Sergey Frolov (Referee 3) · 2023-11-9

Strengths

1 - taken in isolation from the experimental motivation, proposal appears to be solid, though

Weaknesses

1- the paper bases its entire experimental relevance on publications from the same group that present very narrowly selected data and do not offer opportunities for independent verification

Report

The other referees already discussed briefly the theoretical aspects of the proposal. I want to touch on the experimental relevance.

>Our device is inspired by recently reported Cooper pair splitters [15–20]

References 15-20, are from the same group and were not verified by independent experiments. The papers do not provide sufficient data to conclude that they are valid. The group itself has a track record of cherry-picking results, and these papers show very narrowly confined regions of parameter space, such as individual charge degeneracy points - whereas effects are supposed to appear over broad parameter ranges. Having done many experiments in similar regimes, I am surprised by the clarity and definitiveness with which results are presented in those papers. I strongly suspect that if we actually see appropriate amounts of data from those papers the situation will appear very complex and I am not certain that the claims of the papers will hold up.

While this paper only claims to be motivated by these works, some of the authors overlap, and have provided theory for those other works. The paper also centrally features the 'interplay of CAR and ECT' which in fact is the central phenomenon in references 15-20, and lacks proof sufficient for verification by independent experts like myself.

And in the abstract, this paper talks about a "practical implementation" indicating that the relevance of those experiments is paramount.

The concern I have is that we are in a situation of self-verifying research where results are sent into the information space and then taken as valid by the same group. Reminiscent of "Coluomb oscillation" experiments from Copenhagen, where that group published 5 papers all likely unreliable, in top journals, on a technique they themselves invented.

On a larger point, CAR does not appear to be a robust resource for an approach such as quantum computing. This is setting aside the fact that there is no solid evidence that it even exists, though from the viewpoint of basic theory, it should. It is not clear that it should in these wires with a lot of spin relaxation mechanisms present. If I take references 15-20 at their face value, this interplay of CAR and ECT would only happen over extremely narrow ranges of parameters, requiring massive amounts of fine-tuning, and not something you want for your quantum machine.

To add on the subject of unreliable papers, Ref 23 and 30 contain manipulated and selected data, in fact Ref 30 uses data from the same folder as data for the retracted papers 'Quantized Majorana Conductance' and 'Epitaxy of Advanced Nanowire Quantum Devices'. Both papers are from the same institute where this paper is written. Why are the authors feeling the need to refer to unreliable past work?
  • validity: good
  • significance: good
  • originality: good
  • clarity: good
  • formatting: good
  • grammar: perfect

Author:  Kostas Vilkelis  on 2024-04-09  [id 4402]

(in reply to Report 3 by Sergey Frolov on 2023-11-09)

We thank the referee for concluding that our "proposal appears to be solid".

References 15-20, are from the same group and were not verified by independent experiments.

Out of these references, 2 are theoretical, and the 3 experimental works are from 2 different groups. However, upon reviewing our bibliography we realized that it indeed missed multiple important citations, including those to experimental works from other groups. In the updated manuscript we have expanded the bibliography by providing relevant citations.

On a larger point, CAR does not appear to be a robust resource for an approach such as quantum computing. This is setting aside the fact that there is no solid evidence that it even exists, though, from the viewpoint of basic theory, it should. It is not clear that it should in these wires with a lot of spin relaxation mechanisms present.

The referee's claim contradicts multiple works in different systems and from different groups, see our updated bibliography. Therefore we disagree with the referee's claim.

If I take references 15-20 at their face value, this interplay of CAR and ECT would only happen over extremely narrow ranges of parameters, requiring massive amounts of fine-tuning, and not something you want for your quantum machine.

The referee does not provide an explanation in support of this statement, however, we believe every single quantum system requires a massive amount of tuning.

The remainder of the report does not pertain to our work, and therefore we leave it without comment. We believe that with this we addressed all the relevant points raised by the referee.

Attachment:

resubmission_fermionicComputation.pdf

---

## Editorial Decision

resubmitted